# Lookahead Automated Feature Engineering for Tabular Prediction via Kaggle-Guided Knowledge Transfer

**Si-Yang Liu** [1 2 *]   **Zongda Li** [1 2 *]   **Chenming Xu** [1]   **Han Li** [1]   **Ruiqiao Chen** [1]   **Han-Jia Ye** [1 2]

## Abstract

Feature Engineering (FE) can substantially boost accuracy in tabular prediction, yet identifying effective transformations remains challenging: the space of possible operations is vast, and current LLM-based approaches often default to generic "common-sense" features and make myopic, step-by-step choices that overlook beneficial feature combinations. We propose FORGE (**F**eature **O**ptimization with **R**etrieved knowledge **G**uidance and lookahead **E**xploration), a retrieval-augmented lookahead framework that makes LLM-driven feature engineering both more informed and less myopic. FORGE improves the quality of proposed transformations by *transferring* practical feature-engineering expertise: it grounds generation in high-performing Kaggle solutions and retrieves task-relevant patterns that guide the LLM beyond ad hoc transformations. It further improves the search process by enabling lookahead selection over multiple candidate feature programs, increasing the chance of discovering feature sets that work well *together* rather than optimizing each step in isolation. Across real-world tabular tasks with semantically meaningful metadata, FORGE consistently outperforms LLM-based automated feature engineering baselines and serves as a plug-in feature generator whose engineered features generalize across diverse downstream tabular learners.

## 1. Introduction

As learning methods progressed, modern tabular learners, including GBDTs (Chen & Guestrin, 2016; Ke et al., 2017),

deep tabular models (Gorishniy et al., 2021; 2024), and tabular foundation models (Hollmann et al., 2025; Qu et al., 2025), can *implicitly* learn useful representations from data, reducing reliance on manual feature design. However, growing evidence suggests that explicit FE remains a reliable way to improve performance across datasets and model families, and even strong modern models can benefit from well-designed features (Tschalzev et al., 2024). Therefore, automating the discovery of high-value features remains an important and practical goal for tabular machine learning.

Achieving this goal is still challenging. Classical automated feature engineering (AutoFE) methods (Horn et al., 2019; Zhang et al., 2023) typically expand the feature space by applying primitive operators (e.g., arithmetic, aggregation, crossings) and composing them into candidate pipelines. This operator-driven expansion quickly becomes combinatorial: most candidates are redundant or weak, and identifying the few useful ones requires expensive evaluation and careful search. Meanwhile, modern tabular learners may still fail to uncover *task-specific logic* when the predictive signal depends on semantic or relational structure that is not explicit in raw columns (Hollmann et al., 2023b); Appendix C provides an illustrative example. This motivates LLM-based feature engineering, which uses natural-language task descriptions and dataset context to synthesize executable transformation programs that reflect task semantics rather than only statistical regularities.

Recent methods cast LLM-based FE as an agentic generate–evaluate loop (Hollmann et al., 2023b; Nam et al., 2024). At each iteration, an LLM proposes candidate transformation programs based on the task description and the current dataset context; the agent evaluates these proposals on a validation split, accepts or rejects them, updates the feature set, and repeats. This paradigm improves proposal quality, but it leaves two major bottlenecks.

**Trivialization of expertise.** In practice, LLM agents often rediscover a narrow set of "safe" operations (e.g., ratios, simple aggregations, basic log transforms) (Küken et al., 2024), as shown in Figure 1a. Competitive solutions, however, often rely on richer, task-dependent templates such as schema-aware encodings, hierarchy-aware group statistics, and leakage-aware temporal aggregations.

---

*Equal contribution  [1]School of Artificial Intelligence, Nanjing University  [2]National Key Laboratory for Novel Software Technology, Nanjing University. Correspondence to: Han-Jia Ye <yehj@lamda.nju.edu.cn>.

Foundation Models for Structured Data Workshop @ICML 2026, Seoul, South Korea. PMLR 306, 2026. Copyright 2026 by the author(s).

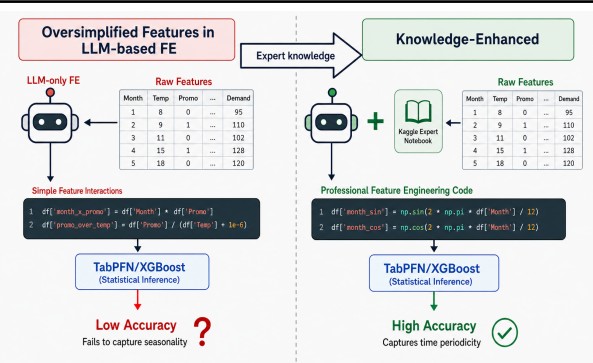

*(a)* Generic proposals without expert grounding.

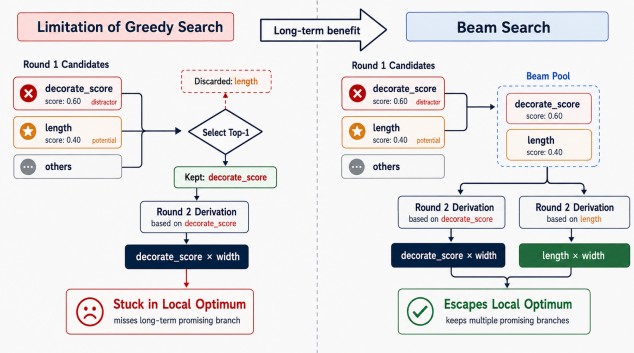

*(b)* Greedy search misses delayed gains.

*Figure 1.* Two bottlenecks in LLM-based FE. **(Left)** Without expert references, LLM agents tend to produce generic, low-value transformations. **(Right)** Stepwise greedy selection can discard intermediate steps whose benefits only appear after composition; maintaining multiple candidates via beam-style search helps discover synergistic feature sets.

**Myopia of stepwise search.** Feature utility is often compositional, so an intermediate transformation may appear weak in isolation while still enabling a high-value feature set after later composition. As a result, a greedy, stepwise strategy tends to discard these enabling steps early because their benefit is delayed, and can therefore miss synergistic feature sets and settle into local optima, as shown in Figure 1b.

To address these issues, we propose **FORGE** (**F**eature **O**ptimization with **R**etrieved knowledge **G**uidance and lookahead **E**xploration), a retrieval-augmented lookahead framework that turns LLM-based FE from myopic refinement into *knowledge-guided, multi-path search*. The core idea is to improve both *the proposals* and *the search* in a single unified loop. First, FORGE performs Kaggle-guided knowledge transfer: it distills high-performing Kaggle solution notebooks into a library of Structured Solution Templates and retrieves task-relevant templates (with lightweight verification) to ground the LLM in proven feature-engineering patterns rather than ad hoc trial-and-error. Second, FORGE introduces lookahead selection by maintaining multiple candidate feature programs and using beam-style search to identify sets of transformations that work well *together*, reducing the risk of discarding intermediate steps whose benefits are delayed. Finally, to keep lookahead exploration practical, FORGE uses TabPFN as a fast search-time proxy to provide low-latency relative feedback during candidate pruning. Viewed together, these components instantiate a three-tier design: expert template retrieval to counter trivialization, multi-trajectory lookahead search to counter myopia, and efficient proxy-based validation to enable broader exploration under realistic budgets.

Our contributions are summarized as follows:

- We propose FORGE, a retrieval-augmented *lookahead* framework for LLM-based FE that mitigates two key failures of current agentic flows: generic feature proposals and myopic stepwise selection.

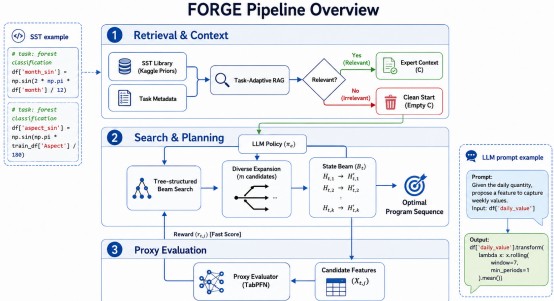

*Figure 2.* The overview of FORGE. It augments the generate–execute–evaluate loop with verified expert references from an SST library, multi-trajectory program search via tree-structured beam search, and low-latency proxy rewards from tabular foundation model.

- FORGE transfers Kaggle feature-engineering know-how via retrieved Structured Solution Templates and enables multi-trajectory lookahead search to capture compositional multi-step feature programs with synergistic gains.
- On semantically described real-world tabular tasks, FORGE consistently improves LLM-based FE baselines and produces features that transfer across diverse downstream tabular learners.

## 2. Methodology

Before presenting the main components of FORGE, we provide the necessary preliminaries and formal problem setup in Appendix A, including the AutoFE setting, the generate–execute–evaluate loop, and the notation used throughout this section. We focus here on the methodological design of FORGE.

### 2.1. Structured Knowledge Injection

FORGE equips the AutoFE loop with transferable expert references to counter the *trivialization of expertise* commonly seen in LLM-based FE. Instead of replaying full

expert pipelines, FORGE retrieves task-relevant **Structured Solution Templates** (SSTs) distilled from expert Kaggle solutions and uses the retained snippets as the expert context $\mathcal{C}$. Given the current task description $\mathcal{T}_{\text{curr}}$, we first retrieve task-similar SST entries and then apply snippet-level utility verification under the current schema summary $\mathcal{S}_0$ to filter out non-transferable code. Conditioned on the verified context $\mathcal{C}$, the LLM proposes higher-impact transformations while remaining grounded in patterns that are both task-relevant and executable under the current schema. Repository construction, retrieval scoring, verifier outputs, and prompt templates are deferred to Appendix C.

### 2.2. Multi-Trajectory Search in Program Space

Even with the expert context $\mathcal{C}$, the space $\Omega^T$ is combinatorial, and feature interactions can exhibit delayed utility: a transformation may yield limited improvement alone, yet become valuable when composed with later operations. A single greedy trajectory is therefore prone to premature commitment and local optima. To overcome stepwise myopia, FORGE explores the combinatorial program space $\Omega^T$ with a tree-structured beam search. At step $t$, each beam state stores the current table, semantic state, and trajectory history, and is prompted with $\mathcal{I} \oplus \mathcal{C} \oplus \mathcal{S}_{t,k} \oplus \mathcal{H}_{t,k}$ to sample candidate programs. The resulting programs are executed, scored by the proxy evaluator, and globally pruned so that only the top-$K$ trajectories survive to the next step. This retain–expand–prune loop preserves intermediate transformations whose benefit emerges only after later composition, reducing the risk of premature commitment to locally attractive but globally weak trajectories. Appendix C provides the full state definition, duplicate-handling rule, history update, and search hyperparameters.

### 2.3. Efficient Search-time Proxy Evaluation

FORGE requires repeatedly scoring candidate feature programs during beam search. Using the final downstream estimator $f_{\text{target}}$ for every candidate is possible in principle, but it can be expensive when the evaluator requires dataset-specific training or hyperparameter tuning, especially because each search round expands up to $K \times m$ candidate states. We therefore instantiate the search-time proxy $f_{\text{proxy}}$ with TabPFN, which supports in-context prediction without per-candidate parameter updates. This choice is motivated primarily by efficiency: TabPFN provides low-latency relative feedback for pruning candidate feature programs, making multi-trajectory search feasible under a practical budget.

We do not claim that TabPFN is the uniquely optimal evaluator for all downstream learners. Rather, it serves as an efficient surrogate during search. When the downstream learner is fixed and sufficient compute is available, using the same learner as the search-time evaluator is a natural

alternative and may yield stronger learner-specific feature choices. In our experiments, we therefore evaluate the selected features not only with TabPFN but also with XGBoost and Random Forest, testing whether proxy-selected transformations transfer beyond the proxy model itself.

Given a fixed train/validation split $(X^{\text{tr}}, y^{\text{tr}}, X^{\text{val}}, y^{\text{val}})$, for a candidate state $X'$, TabPFN produces

$$\hat{y}^{\text{val}} = f_{\text{proxy}}(X'^{\text{tr}}, y^{\text{tr}}, X'^{\text{val}}).$$

We set $V$ to ROC-AUC for classification and negative RMSE for regression, and compute the search-time reward as

$$r = V(\hat{y}^{\text{val}}, y^{\text{val}}).$$

The reward is used only for candidate ranking and pruning during search; final performance is always reported using held-out evaluation with the downstream learners described in Section 3.2.

## 3. Experiments

### 3.1. Datasets

We evaluate FORGE on real-world tabular prediction tasks with semantically informative column names and task descriptions. This setting is intentional: FORGE targets semantic LLM-based feature engineering, where column meanings, target semantics, and task context are part of the input signal. Fully anonymized tables remove this signal and correspond to a different problem setting closer to operator search over anonymous variables. We provide the detailed filtering protocol and dataset statistics in Appendix D.

### 3.2. Baselines and Evaluation Protocols

**Downstream evaluators and transferability.** We evaluate the constructed features with Random Forest, XGBoost, and TabPFN. During FORGE search, TabPFN is used as the only search-time proxy evaluator to select the transformation program $\Phi^*$. [1] The selected $\Phi^*$ is then applied to the data and evaluated with all three downstream learners, testing whether proxy-selected features transfer beyond the proxy model itself.

**Baselines.** We compare with (i) raw features with minimal preprocessing, including imputation and ordinal encoding; (ii) classical AutoFE methods, including AutoFeat (Horn et al., 2019) and OpenFE (Zhang et al., 2023); and (iii) LLM-based FE agents, including CAAFE (Hollmann et al., 2023b) and OCTree (Nam et al., 2024). All LLM-based methods use the same LLM backbone and evaluation pipeline whenever applicable.

---

[1] This design prioritizes low-latency candidate ranking rather than claiming TabPFN to be the optimal evaluator for every downstream learner.

*Table 1.* **Main results** aggregated over 23 datasets under three downstream evaluators (TabPFN, XGBoost, Random Forest). **Avg.** denotes the mean across datasets and **Med.** denotes the median across datasets. **Avg. Rank** (↓): lower is better. **Rel. Improv.**: relative improvement over the *No FE* baseline (Original). **Wins vs. No FE**: number of datasets where the method outperforms the raw-feature baseline (Original).

| | Baselines | | | CAAFE (Hollmann et al., 2023b) | | OCTree (Nam et al., 2024) | |
|---|---|---|---|---|---|---|---|
| **Metric** | **Original** | **AutoFE** | **OpenFE** | **Vanilla** | **+ FORGE (Ours)** | **Vanilla** | **+ FORGE (Ours)** |
| **Downstream Evaluator: TabPFN** | | | | | | | |
| Avg. Rank (↓) | 6.57 | 6.22 | 6.30 | 4.83 | **1.78** | 5.17 | 2.65 |
| Avg. Rel. Imp. (↑) | - | +0.03% | +0.02% | +0.38% | **+1.95%** | +0.26% | +0.87% |
| Med. Rel. Imp. (↑) | - | +0.00% | +0.00% | +0.00% | **+0.81%** | +0.00% | +0.20% |
| Wins vs. No FE (Count) | - | 3/23 | 1/23 | 11/23 | **22/23** | 8/23 | 21/23 |
| **Downstream Evaluator: XGBoost** | | | | | | | |
| Avg. Rank (↓) | 6.61 | 6.30 | 6.48 | 4.57 | **2.13** | 5.48 | 2.70 |
| Avg. Rel. Imp. (↑) | - | +0.08% | +0.00% | +0.29% | **+1.63%** | -0.07% | +1.28% |
| Med. Rel. Imp. (↑) | - | +0.00% | +0.00% | +0.00% | **+0.40%** | +0.00% | +0.09% |
| Wins vs. No FE (Count) | - | 2/23 | 1/23 | 12/23 | **21/23** | 7/23 | 19/23 |
| **Downstream Evaluator: Random Forest** | | | | | | | |
| Avg. Rank (↓) | 6.35 | 5.48 | 5.96 | 5.57 | 2.43 | 5.13 | **2.26** |
| Avg. Rel. Imp. (↑) | - | +0.32% | +0.04% | +0.08% | +1.23% | +0.36% | **+1.29%** |
| Med. Rel. Imp. (↑) | - | +0.00% | +0.00% | +0.00% | **+0.52%** | +0.00% | +0.14% |
| Wins vs. No FE (Count) | - | 5/23 | 2/23 | 8/23 | 20/23 | 6/23 | **21/23** |

### 3.3. Experimental Setup

For datasets with official labeled splits, we preserve the original split when available; otherwise, we use five repeated random train/test splits with fixed seeds, stratified for classification. Candidate feature programs are selected on an internal validation split, and final performance is measured on held-out data. All feature transformations are fitted using training data only and then applied to validation/test data to avoid target leakage.

We report ROC-AUC for classification and RMSE for regression. Aggregate comparisons use average rank, relative improvement over the raw-feature baseline, and win counts. Unless otherwise specified, all LLM-based methods use `DeepSeek-V3.2` with sampling temperature $\tau = 0.7$. All iterative methods run for at most 10 rounds. For FORGE, we set beam width $K = 3$ and expansion width $m = 3$; invalid programs are rejected during execution.

### 3.4. Main Results

Table 1 summarizes results on 23 datasets and evaluates FORGE in terms of effectiveness, plug-in extensibility, and cross-model transferability.

**Effectiveness.** Across evaluators, FORGE achieves the best overall performance among the compared methods. Under the TabPFN evaluator (top block), FORGE+CAAFE attains the best Avg. Rank (**1.78**), outperforming traditional AutoFE baselines (e.g., AutoFE, Avg. Rank 6.22) and vanilla LLM agents (e.g., Vanilla CAAFE, Avg. Rank 4.83).

In dataset-wise comparisons against the *No FE* baseline (Original), FORGE+CAAFE improves performance on **22/23** datasets, indicating that expert priors combined with multi-trajectory search can uncover useful feature interactions missed by greedy baselines.

To disentangle the contribution of each module and assess robustness, we conduct additional analyses on a subset of **6 datasets** (3 classification and 3 regression). Ablations in Appendix E show that both expert priors and lookahead search contribute to the final gains, and expert priors are most effective when coupled with lookahead search. We also defer the semantic-blindness analysis, utility-verification study, and qualitative code examples to Appendix E.

## 4. Conclusion

In this work, we introduced **FORGE**, a retrieval-augmented lookahead framework for LLM-based automated feature engineering. FORGE combines retrieved expert templates distilled from top Kaggle solutions, multi-trajectory lookahead search for composing feature programs, and low-latency proxy evaluation via TabPFN. Across diverse real-world tabular datasets, FORGE consistently improves strong AutoFE baselines and acts as a plug-and-play feature generator that benefits a range of downstream learners. These results support the "Feature Engineering by Proxy" paradigm, highlighting tabular foundation models as practical evaluators for scalable, knowledge-guided feature discovery.

# Acknowledgements

This work is partially supported by National Key R&D Program of China (2024YFE0202800).

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

The Appendix consists of eight sections:

1. Appendix A: We provide the problem setup and the formal *generate–execute–evaluate* view of LLM-based feature engineering.
2. Appendix B: We review prior studies on automated feature engineering, LLM-based agents for program synthesis, and tabular foundation models, positioning FORGE in the broader landscape.
3. Appendix C: We provide the full experimental setup, including prompting templates and downstream model details.
4. Appendix D: We summarize the datasets used in our experiments and the construction of the SST repository.
5. Appendix E: We report extended quantitative results that are omitted from the main text due to space constraints.
6. Appendix F: We discuss the current limitations of FORGE and outline promising directions for future extensions.
7. Appendix G: We provide the declaration of LLM usage.
8. Appendix H: We acknowledge the Kaggle platform and the original authors of the public notebooks, solutions, and datasets used to construct the SST repository, and describe our attribution and licensing policy.

## A. Preliminaries

In this section, we introduce the basic setup of tabular prediction and automated feature engineering, and formalize LLM-based FE as a sequential *generate–execute–evaluate* loop. We define the heterogeneous tabular representation, the semantic context provided to the LLM, and the evaluation objective that scores candidate transformations.

### A.1. Tabular Prediction and Implicit Feature Extraction

We consider supervised tabular prediction with dataset $\mathcal{D} = (\mathbf{X}, \mathbf{y})$, where $\mathbf{X} \in \mathcal{X}_1 \times \cdots \times \mathcal{X}_M$ contains $M$ heterogeneous columns (e.g., numerical, categorical, or time-related), and $\mathbf{y} \in \mathcal{Y}^N$ denotes the target labels for $N$ instances. We focus on standard settings where $\mathcal{Y} = \{0, 1\}$ for binary classification, $\mathcal{Y} = \{1, \ldots, C\}$ for multi-class classification, and $\mathcal{Y} = \mathbb{R}$ for regression.

Modern tabular learners (e.g., tabular foundation models) can be viewed as implicitly performing representation learning, which we write as

$$f(\mathbf{X}) = h\big(g(\mathbf{X})\big), \tag{1}$$

where $g(\cdot)$ maps the raw table to learned representations and $h(\cdot)$ is a prediction head. Despite strong implicit feature extraction, predictive signals in real-world tasks may depend on semantic or relational structure that is not explicit in raw columns, making explicit feature transformations still important in practice (Tschalzev et al., 2024).

### A.2. LLM-Based FE

**Classical AutoFE.** A common way to obtain explicit transformations is automated feature engineering (AutoFE), which expands the feature space by enumerating and composing predefined operators (e.g., arithmetic, crossing, and aggregation), and selects useful candidates through evaluation. Given an operator library $\Omega$, this process can be abstracted as constructing a candidate pool

$$\mathcal{Z}(\mathbf{X}) = \{\phi(\mathbf{X}) \mid \phi \in \Omega\}, \tag{2}$$

followed by selecting a small subset that improves validation performance. While effective in principle, the candidate space grows rapidly with operator compositions and is largely blind to task semantics, making efficient search and evaluation a key bottleneck in practice. These limitations motivate LLM-based FE, which injects semantic guidance by generating executable transformation programs conditioned on task descriptions and dataset context.

LLM-based FE treats feature construction as *program synthesis* guided by semantic context, and improves feature sets through an iterative *generate–execute–evaluate* loop. At iteration $t$, the agent produces an executable transformation program $\phi_t$ from a valid operator space $\Omega$, applies it to the current table, and evaluates the updated features on held-out data to obtain feedback for the next iteration.

**LLM operator.** We denote the LLM as a conditional program generator $\mathcal{M}_\theta$. Given a structured prompt context $\mathcal{P}_t$, the LLM samples an executable transformation program:

$$\phi_t \sim \mathcal{M}_\theta(\cdot \mid \mathcal{P}_t), \qquad \mathcal{P}_t = \mathcal{I} \oplus \mathcal{C} \oplus \mathcal{S}_t \oplus \mathcal{H}_t. \tag{3}$$

Here, $\mathcal{I}$ specifies system-level constraints (e.g., Python/Pandas format and execution rules), $\mathcal{C}$ provides optional external references (e.g., retrieved expert examples), $\mathcal{S}_t$ summarizes the currently available features, and $\mathcal{H}_t$ records previous actions and feedback. The output $\phi_t \in \Omega$ is a code snippet that implements a feature transformation, such as arithmetic combinations, group-wise aggregations, encodings, or time-based operations.

**Semantic state.** We define the semantic state $\mathcal{S}_t$ as the structured dataset view exposed to the LLM at iteration $t$. Initially, $\mathcal{S}_0$ summarizes the raw schema as

$$\mathcal{S}_0 = \{(h_j, \tau_j, \mathbf{v}_j)\}_{j=1}^M \cup \{\mathcal{T}_{\text{task}}\}, \tag{4}$$

where $h_j$ denotes the feature name, $\tau_j \in \{\texttt{Num}, \texttt{Cat}, \texttt{Time}\}$ denotes the feature type, and $\mathbf{v}_j$ contains representative values (e.g., the first $K$ non-null entries). $\mathcal{T}_{\text{task}}$ is a natural-language description of the prediction target. After each transformation, $\mathcal{S}_t$ is updated to reflect the current feature set, thereby constraining the operands and types available for subsequent programs.

**Execution.** Given the generated program $\phi_t$, we apply it to the current feature table via execution:

$$\mathbf{X}_{t+1} = \texttt{Exec}(\mathbf{X}_t, \phi_t). \tag{5}$$

A transformation is considered valid if it is executable under $\mathcal{I}$ and compatible with the operands described in $\mathcal{S}_t$. If execution fails (e.g., type mismatch or runtime error), the candidate can be rejected by assigning a low reward.

**Evaluation and feedback.** A candidate transformation is useful if it improves predictive performance on held-out data. The ideal objective is to maximize the downstream performance of a target estimator $f_{\text{target}}$:

$$\Phi^* = \arg \max_{\Phi \in \Omega^T} \mathcal{V}\Big(f_{\text{target}}(\texttt{Exec}(\mathbf{X}_0, \Phi)), \mathbf{y}\Big), \tag{6}$$

where $\Phi = (\phi_1, \ldots, \phi_T)$ and $\mathcal{V}$ denotes a validation metric. However, repeatedly training $f_{\text{target}}$ for every candidate is expensive during search. Therefore, LLM-based FE typically relies on a fast proxy evaluator $f_{\text{proxy}}$ to provide step-wise feedback.

Given a fixed train/validation split $(\mathbf{X}^{\text{tr}}, \mathbf{y}^{\text{tr}}, \mathbf{X}^{\text{val}}, \mathbf{y}^{\text{val}})$, we define a proxy mapping

$$f_{\text{proxy}} : (\mathbf{X}^{\text{tr}}, \mathbf{y}^{\text{tr}}, \mathbf{X}^{\text{val}}) \mapsto \hat{\mathbf{y}}^{\text{val}}, \tag{7}$$

and compute the reward for the updated dataset as

$$r_t = \mathcal{V}(\hat{\mathbf{y}}_{t+1}^{\text{val}}, \mathbf{y}^{\text{val}}), \qquad \hat{\mathbf{y}}_{t+1}^{\text{val}} = f_{\text{proxy}}(\mathbf{X}_{t+1}^{\text{tr}}, \mathbf{y}^{\text{tr}}, \mathbf{X}_{t+1}^{\text{val}}). \tag{8}$$

Finally, the history buffer is updated with the new observation:

$$\mathcal{H}_{t+1} = \mathcal{H}_t \oplus (\phi_t, r_t), \tag{9}$$

which provides feedback for subsequent iterations. In FORGE, we instantiate $f_{\text{proxy}}$ with TabPFN, which produces $\hat{\mathbf{y}}^{\text{val}}$ via in-context prediction without retraining, enabling low-latency statistical feedback within the loop.

### A.3. A Short Summary

Overall, automated feature engineering can be viewed as searching for executable transformation programs under a compute budget. Classical AutoFE expands candidate features by composing primitive operators, but often suffers from combinatorial explosion and expensive evaluation. LLM-based FE introduces semantic guidance by generating transformation programs from task descriptions and dataset context, and refines them through the *generate–execute–evaluate* loop described above. This unified view motivates approaches that can incorporate transferable expert references and obtain fast statistical feedback, enabling scalable exploration of feature transformations. A detailed related work is presented in Appendix B

## B. Related Work

### B.1. Tabular Representation Learning

Tabular data, distinguished by its heterogeneous feature spaces and non-smooth manifolds, presents unique challenges for representation learning. For decades, Gradient Boosted Decision Trees (GBDTs), such as XGBoost (Chen & Guestrin, 2016),

LightGBM (Ke et al., 2017), and CatBoost (Prokhorenkova et al., 2018), have remained the dominant choice. Empirical studies (Grinsztajn et al., 2022) attribute their dominance to inductive biases that favor irregular decision boundaries (Beyazit et al., 2023), which align naturally with tabular distributions. While Deep Tabular Learning (DTL) architectures like FT-Transformer (Gorishniy et al., 2021), ModernNCA (Ye et al., 2025b), and TabM (Gorishniy et al., 2024) have attempted to introduce end-to-end differentiability, they frequently fail to consistently outperform well-tuned GBDTs on small-to-medium datasets without extensive regularization (Grinsztajn et al., 2022; McElfresh et al., 2023; Ye et al., 2024; Jiang et al., 2025a). Recently, a paradigm shift has occurred with the advent of Tabular Foundation Models (TFMs) (Ye et al., 2025a; Liu & Ye, 2025). Models such as TabPFN (Hollmann et al., 2023a) and its successors (e.g., TabPFN v2 (Hollmann et al., 2025), TabICL (Qu et al., 2025)), and Mitra (Zhang et al., 2025b) utilize Transformers pre-trained on massive synthetic priors to execute in-context learning (ICL). By approximating Bayesian inference via a single forward pass, these models achieve state-of-the-art zero-shot performance. However, despite their statistical proficiency, current TFMs predominantly process features as anonymous numerical vectors. This design renders them *semantically blind*: while adept at modeling statistical dependencies, they lack the *semantic reasoning capabilities* to interpret the domain metadata governing the data. This limitation underscores the critical need for external semantic reasoning to bridge the gap between raw statistical representation and logical meaning.

## B.2. LLMs for Enhancing Machine Learning Pipelines

Despite the strong empirical success of machine learning (ML), building high-performing ML pipelines remains difficult due to the large space of design choices spanning data preparation, feature construction, model selection, and hyperparameter tuning. AutoML (Hutter et al., 2019) seeks to reduce this burden via techniques such as neural architecture search (Pham et al., 2018) and Bayesian optimization (Frazier, 2018). However, many AutoML systems can be computationally expensive, may transfer poorly across tasks, and often provide limited interpretability into why a particular pipeline works (Zhang et al., 2024). Motivated by the broad capabilities of Large Language Models (LLMs), recent work has begun to use LLMs and LLM-based agents to assist different stages of the ML workflow. Prior studies have explored LLM support for task understanding and problem formulation (Pricope, 2025; Chan et al., 2025), data cleaning (Bendinelli et al., 2025; Bodensohn et al., 2025), feature engineering (Hollmann et al., 2023b; Nam et al., 2024), and model building and tuning (Li et al., 2024; Zhang et al., 2024; Jiang et al., 2025b; Liu et al., 2025b). A common dependency in these approaches is access to semantic side information, such as column descriptions, dataset metadata, or natural-language documentation, which can strongly influence generation quality and downstream reliability.

## B.3. LLM-Driven Automated Feature Engineering

The domain of LLM-driven feature engineering, pioneered by **CAAFE** (Hollmann et al., 2023b), frames feature generation as a semantic optimization task rather than a purely statistical one. Recent advancements in this field can be categorized into three distinct methodological paradigms:

**Iterative and Evolutionary Optimization.** The foundational paradigm employs a closed-loop feedback mechanism. **CAAFE** established the "generate-evaluate-refine" loop, employing **TabPFN v1** (restricted to classification tasks) to greedily filter generated features based on validation scores. **ReFeat** (Han et al., 2025) enhances this by introducing a recursive elimination mechanism to prune redundant features. Most recently, **LLM-FE** (Abhyankar et al., 2025) models feature generation as an *Evolutionary Search*. It maintains a population of feature programs and utilizes the LLM as a mutation operator.

**Reasoning and Knowledge Augmentation.** Recognizing that performance signals alone are insufficient, recent works integrate explicit reasoning strategies. **FeatLLM** (Han et al., 2024) employs Chain-of-Thought (CoT) prompting to induce explicit logical rules (e.g., IF-THEN clauses) for few-shot scenarios. **ReFeat** (Han et al., 2025) introduces a meta-cognitive layer, modeling the choice of reasoning strategy (e.g., inductive vs. deductive) as a Multi-Armed Bandit (MAB) problem. In terms of external knowledge, **ReAGen** (Bouadi et al., 2025) utilizes Retrieval-Augmented Generation to query Domain Knowledge Graphs (e.g., medical ontologies). However, ReAGen retrieves *domain facts* (symbolic relations) rather than *feature engineering methodologies* (computational logic).

**Structure and Alignment Guidance.** A parallel trend leverages intrinsic model structures or alignment techniques to guide generation. **OCTree** (Nam et al., 2024) extracts decision paths from tree-based models and translates them into semantic prompts, explicitly guiding the LLM to formalize non-linear interactions captured by the tree. **FeRG-LLM** (Ko et al., 2025) adopts a fine-tuning approach, using Direct Preference Optimization (DPO) to align a smaller LLM (Llama-3-8B) with the

*Table 2.* Statistics for the Forest Cover Type dataset.

| Dataset | Samples | Features | Classes |
|---|---|---|---|
| Forest Cover Type (Kernels Only) | 15,120 | 54 | 7 |

utility preferences of feature engineering tasks, implicitly internalizing expert intuition.

While distinct progress has been made, a unified framework remains elusive.

- **Search Myopia:** Methods like CAAFE and LLM-FE either rely on greedy search or limit their "memory" to internal trial-and-error, lacking the global perspective of *crystallized expert solutions*;
- **Knowledge Gap:** While ReaGen introduces RAG, it focuses on declarative domain knowledge ("What is Creatinine?") rather than procedural expert logic ("How do Grandmasters aggregate time-series?").

### B.4. Reasoning Augmentation via Retrieval and Search

To mitigate the limitations of triviality and myopia, we draw inspiration from advancements in code reasoning: Retrieval-Augmented Generation (RAG) and Tree Search. Studies such as DocPrompting (Zhou et al., 2023) and CodeRAG (Zhang et al., 2025a) demonstrate that retrieving relevant documentation significantly enhances generation correctness. In tabular FE, this translates to the necessity of retrieving proven feature engineering "recipes" rather than generating them *ex nihilo*. Furthermore, complex reasoning tasks have evolved from linear Chain-of-Thought (CoT) (Wei et al., 2022) to tree-based planning. Algorithms like Tree of Thoughts (ToT) (Yao et al., 2023) and LATS (Zhou et al., 2024) enable lookahead and backtracking. Notably, SELA (Chi et al., 2024) introduced Monte Carlo Tree Search (MCTS) to AutoML pipeline search. However, a significant impediment to adopting tree search in Feature Engineering is the prohibitive computational cost of evaluating each candidate state. FORGE circumvents this barrier by exploiting the inference efficiency of TFMs as a low-latency proxy evaluator. This renders Beam Search in the feature program space computationally feasible, effectively synergizing deep expert retrieval with global search.

## C. Implementation Details

### C.1. SST Library and Retrieval

#### C.1.1. AN EXAMPLE OF A STORED SST

Table 2 shows an example SST entry from our SST repository.

**Dataset overview.** The dataset contains cartographic variables (e.g., elevation, slope, distances to hydrology/roadways, and hillshade indices) and the goal is to classify forest cover types.

**Example feature descriptions.** Below we list representative feature names and their meanings (abridged for readability):

- `Aspect`: Aspect in degrees azimuth.
- `Slope`: Slope in degrees.
- `HorizontalDistanceToHydrology`: Horizontal distance to the nearest surface water feature.
- `VerticalDistanceToHydrology`: Vertical distance to the nearest surface water feature.
- `HorizontalDistanceToRoadways`: Horizontal distance to the nearest roadway.
- `Hillshade9am`: Hillshade index at 9am on the summer solstice.
- `HillshadeNoon`: Hillshade index at noon on the summer solstice.
- `Hillshade3pm`: Hillshade index at 3pm on the summer solstice.
- `HorizontalDistanceToFirePoints`: Horizontal distance to the nearest wildfire ignition point.
- `WildernessArea`: Wilderness area designation.
- `SoilType`: Soil type designation.
- `CoverType`: Target label (forest cover type).

The following code snippet shows one representative feature-engineering template extracted from the associated notebook.

```python
def main(train_df, test_df):
    type_ratio = np.array([0.37053, 0.49681, 0.05936, 0.00103, 0.01295, 0.02687, 0.03242])

    total_df = pd.concat([train_df.iloc[:, :-1], test_df])

    total_df["Aspect_Sin"] = np.sin(np.pi*total_df["Aspect"]/180)
    total_df["Aspect_Cos"] = np.cos(np.pi*total_df["Aspect"]/180)

    hillshade_col = ["Hillshade_9am", "Hillshade_Noon", "Hillshade_3pm"]
    for col1, col2 in combinations(hillshade_col, 2):
        total_df[col1 + "_add_" + col2] = total_df[col2] + total_df[col1]
        total_df[col1 + "_dif_" + col2] = total_df[col2] - total_df[col1]
        total_df[col1 + "_div_" + col2] = (total_df[col2]+0.01) / (total_df[col1]+0.01)
        total_df[col1 + "_abs_" + col2] = np.abs(total_df[col2] - total_df[col1])

    total_df["Hillshade_mean"] = total_df[hillshade_col].mean(axis=1)
    total_df["Hillshade_std"] = total_df[hillshade_col].std(axis=1)
    total_df["Hillshade_max"] = total_df[hillshade_col].max(axis=1)
    total_df["Hillshade_min"] = total_df[hillshade_col].min(axis=1)

    total_df["Degree_to_Hydrology"] = ((total_df["Vertical_Distance_To_Hydrology"] +
        0.001) /
    (total_df["Horizontal_Distance_To_Hydrology"] + 0.01))

    horizontal_col = ["Horizontal_Distance_To_Hydrology",
                      "Horizontal_Distance_To_Roadways",
                      "Horizontal_Distance_To_Fire_Points"]

    for col1, col2 in combinations(hillshade_col, 2):
        total_df[col1 + "_add_" + col2] = total_df[col2] + total_df[col1]
        total_df[col1 + "_dif_" + col2] = total_df[col2] - total_df[col1]
        total_df[col1 + "_div_" + col2] = (total_df[col2]+0.01) / (total_df[col1]+0.01)
        total_df[col1 + "_abs_" + col2] = np.abs(total_df[col2] - total_df[col1])
    return total_df
```

*Listing 1.* Feature engineering for Forest Cover Type (Kernels Only)

### C.1.2. REPOSITORY CONSTRUCTION AND ENTRY FORMAT

We build an SST repository by curating public top-ranking Kaggle solutions through a standardization pipeline. The pipeline consists of three steps. *(1) Competition filtering.* We collect 90 diverse tabular competitions with rich semantic descriptions and finalized leaderboards, prioritizing tasks where non-trivial feature interactions are crucial. *(2) Solution selection.* For each competition, we retain the *top-3* solutions that contain explicit feature engineering logic, while filtering out notebooks dominated by ensembling or hyperparameter tuning. *(3) Asset standardization and decoupling.* We decouple feature transformations from dataset-specific preprocessing (e.g., hard-coded cleaning rules) and model training loops, retaining only self-contained and reusable feature programs.

As shown in Section C.1.1, each SST entry is represented as $\tau = (\mathcal{M}_{task}, \mathcal{C}_{code})$. $\mathcal{M}_{task}$ provides semantic metadata for retrieval (e.g., dataset overview and feature meanings), and $\mathcal{C}_{code}$ contains executable feature-engineering templates extracted from expert notebooks. Overall, this process yields a curated SST library (top-3 solutions across 90 competitions), which is retrieved (with lightweight verification) to construct $\mathcal{C}$, grounding FORGE's proposals and steering its lookahead search toward high-value regions of the transformation space.

### C.1.3. THE DETAILS OF THE RETRIEVAL PROCESS

**Retrieval with Semantic Utility Verification.** To reduce generic feature proposals, FORGE retrieves expert feature-engineering templates from task-similar SST entries and uses them as in-context guidance for proposal generation and lookahead composition. Given the current task description $\mathcal{T}_{curr}$, we encode it into a dense vector using a language-model embedding encoder $\mathbf{e}(\cdot)$. For each SST entry, we similarly embed its metadata $\mathcal{M}_{task}$ and retrieve the most similar entry by cosine similarity in the embedding space.

However, task similarity does not guarantee code applicability: even for similar tasks, a template may rely on missing fields, incompatible types, or dataset-specific assumptions, which can introduce negative transfer. FORGE therefore follows a *retrieve–then–verify* design.

**Snippet-level utility verification.** Each SST entry contains three feature-engineering templates, and we verify them individually by asking an LLM whether the template is usable under the current schema summary $\mathcal{S}_0$ (example prompt in Appendix C). We denote the utility decision as

$$u(\mathcal{C}_{code}^{(i)}, \mathcal{S}_0) \in \{0, 1\}, \qquad i \in \{1, 2, 3\}, \tag{10}$$

where $u = 1$ indicates that the template can be executed (or adapted with minimal column-name renaming) using available columns and type-compatible operations described in $\mathcal{S}_0$, while $u = 0$ indicates missing operands or non-transferable assumptions. The retained templates form the final expert reference context

$$\mathcal{C} = \{\mathcal{C}_{code}^{(i)} \mid u(\mathcal{C}_{code}^{(i)}, \mathcal{S}_0) = 1, \ i \in \{1, 2, 3\}\}, \tag{11}$$

which is injected into FORGE's proposal generator and lookahead search stage as in-context exemplars.

### C.1.4. PROMPT TEMPLATE FOR SNIPPET-LEVEL UTILITY VERIFICATION

```
## Task Description
You are a transferability verifier in a Forge retrieval pipeline. Given the current task
    description and a retrieved expert notebook, decide whether the reference should be
    discarded to avoid negative transfer.

## Decision Rule
Output 1 (ACCEPT) if the retrieved reference is likely reusable as an in-context expert
    hint.
Output 0 (REJECT) if the retrieved reference is unlikely to transfer safely.

## Rejection Criteria (any one is enough for REJECT=0)
1. Schema mismatch: relies on columns/fields that are missing in the current dataset.
2. Dataset-specific assumptions: depends on source-specific IDs, naming conventions, or
    domain rules that are not generalizable.
3. Task mismatch: method is designed for a different prediction objective, target
    semantics, or learning setting.
4. Fragile transfer: requires preprocessing or metadata not available in the current
    context.

## Input Data
### CURRENT TASK / DATASET DESCRIPTION:
{description}

##dataset samples:
{samples}

## RETRIEVED EXPERT REFERENCE:
{expert feature engineering code}

## Output Requirements
- Return exactly one token: 1 or 0
- 1 means accept, 0 means reject
- No explanation, no extra text, no punctuation, no quotation marks
FAILURE TO FOLLOW OUTPUT FORMAT WILL RESULT IN SYSTEM ERROR
```

*Listing 2.* Prompt template for snippet-level utility verification

We provide a prompt template for snippet-level utility verification in Listing 2.

## C.2. Multi-Trajectory Search in Program Space

### C.2.1. STATE REPRESENTATION AND BEAM TRANSITION

**State representation.** At step $t$, the $k$-th beam element is

$$S_{t,k} = (\mathbf{X}_{t,k}, \mathcal{S}_{t,k}, \mathcal{H}_{t,k}), \tag{12}$$

where $\mathbf{X}_{t,k}$ is the current table state, $\mathcal{S}_{t,k}$ is the semantic state exposed to the LLM (the schema/feature summary defined in Section A), and $\mathcal{H}_{t,k}$ records the trajectory-specific history of programs and rewards. We initialize $\mathcal{S}_0$ as the schema summary of the original table and update it to $\mathcal{S}_{t,k}$ after each execution. We denote the beam as $\mathcal{B}_t = \{S_{t,1}, \ldots, S_{t,K}\}$.

**Beam transition.** Each iteration applies an expand–evaluate–prune procedure.

**(1) Diverse expansion.** For each $S_{t,k}$, we construct $\mathcal{P}_{t,k} = \mathcal{I} \oplus \mathcal{C} \oplus \mathcal{S}_{t,k} \oplus \mathcal{H}_{t,k}$ where $\mathcal{C}$ is prepended as a fixed expert-reference block for every trajectory, and $\mathcal{S}_{t,k}$ and $\mathcal{H}_{t,k}$ provide trajectory-specific context. We then sample $m$ programs from $\mathcal{M}_\theta(\cdot \mid \mathcal{P}_{t,k})$ using temperature sampling to encourage diverse operator choices. We discard duplicate programs (e.g., identical normalized code strings) to avoid redundant expansions. This produces up to $N = K \times m$ candidates.

**(2) Execution and proxy evaluation.** Each candidate program $\phi_{k,j}$ is executed to obtain $\mathbf{X}'_{k,j} = \texttt{Exec}(\mathbf{X}_{t,k}, \phi_{k,j})$ and scored by FORGE's proxy evaluator (TabPFN) to produce reward $r_{k,j}$. Invalid programs (syntax errors or runtime failures) are assigned a reward of $-\infty$.

**(3) Global top-$K$ pruning with trajectory-specific updates.** We rank all candidates by their proxy rewards $r_{k,j}$ and keep the top-$K$ survivors to form $\mathcal{B}_{t+1}$, where $r_{k,j}$ reflects the trajectory reward accumulated up to the current step. For each survivor indexed by $k'$, the history is updated as

$$\mathcal{H}_{t+1,k'} = \mathcal{H}_{t,k} \oplus (\phi_{k,j}, r_{k,j}). \tag{13}$$

Maintaining multiple trajectories reduces the risk of discarding enabling transformations too early and supports discovering synergistic feature interactions through composition.

### C.2.2. PROMPTING TEMPLATE FOR PROGRAM SYNTHESIS

Figure 3 shows an example of the prompt in the main program.

### C.2.3. DETAILS OF THE BEAM SEARCH

In the main experiment and ablation experiment, we set the beam width to $K = 3$ and expand each beam with $m = 3$ candidates per step. The maximum number of iterative rounds is $n = 10$ with number of initialization candidates set to $N_{init} = 5$. The beam score for ranking the children is $-$rmse for regression tasks, roc-auc for classification tasks, and $-inf$ for errors in evaluation. We apply dynamic prompt construction in the feedback process. We implement two distinct modes using Python string templates:

**Debugging Mode:** Triggered when a candidate code fails to execute. The system sends the error and constructs a prompt "analyze the logic, fix the bug, and provide the complete corrected code".

**Optimization Mode:** Triggered when the code executes successfully. The system serializes the runtime metrics (roc and acc for classification, rmse for regression) into a JSON string. The metrics are sent with the prompt "analyze these metrics and optimize the feature engineering or model parameters to achieve a higher score".

## C.3. Details of the downstream models

In this section, we provide detailed specifications of the downstream models involved in our evaluation.

**TabPFN.** We utilize TabPFN (Grinsztajn et al., 2025), a Transformer-based tabular foundation model. Unlike traditional models that require optimization per dataset, TabPFN employs in-context learning to generate predictions from a set of training examples within a single forward pass. We use the official `tabpfn` PyTorch implementation with the standard pre-trained checkpoint and do not perform additional fine-tuning. We set $N_{ensemble} = 1$ to accelerate inference. Due to the model's context window constraint, we subsample the training data to 5000 samples for datasets exceeding this limit, while using the full validation/test sets for evaluation.

*Table 3.* Hyperparameter search spaces for downstream tree-based evaluators. We follow the search spaces adopted by TALENT (Liu et al., 2025a) and run 100 trials for each evaluation setting.

| Model | Hyperparameter | Search space |
|---|---|---|
| XGBoost | alpha | optional zero or LogUniform$(10^{-8}, 100)$ |
| XGBoost | gamma | optional zero or LogUniform$(10^{-8}, 100)$ |
| XGBoost | lambda | optional zero or LogUniform$(10^{-8}, 100)$ |
| XGBoost | learning_rate | LogUniform$(10^{-5}, 1)$ |
| XGBoost | max_depth | Int$(3, 10)$ |
| XGBoost | min_child_weight | LogUniform$(10^{-8}, 10^5)$ |
| XGBoost | colsample_bylevel | Uniform$(0.5, 1.0)$ |
| XGBoost | colsample_bytree | Uniform$(0.5, 1.0)$ |
| XGBoost | subsample | Uniform$(0.5, 1.0)$ |
| Random Forest | min_samples_split | Int$(2, 10)$ |
| Random Forest | min_samples_leaf | Int$(1, 10)$ |

**XGBoost and Random Forest.** For XGBoost (Chen & Guestrin, 2016), we use the `xgboost` library (Ver. 3.1.3). For Random Forest (Breiman, 2001), we use `RandomForestClassifier` and `RandomForestRegressor` from scikit-learn (Ver. 1.7.2). Instead of using default hyperparameters, we follow the hyperparameter search spaces adopted by TALENT (Liu et al., 2025a) and perform 100 trials for each downstream evaluation setting. The search spaces are summarized in Table 3. All other fitting options are kept unchanged.

### C.4. Compute and Reproducibility

We conducted all experiments on a machine equipped with two Intel Xeon Platinum 8488C CPUs and four NVIDIA RTX 6000 Ada Generation GPUs. To make the practical cost of FORGE more transparent, we further break down the average runtime of one search iteration into its main components. On average, the LLM-calling time is 68.52s, and the TabPFN evaluator time for a single candidate evaluation is 3.64s. For final downstream training and testing, the average runtime is 2.89s for TabPFN and 25.95s for XGBoost.

We also report the average token usage of the LLM calls. Across datasets, each prompt contains 6,187 tokens on average, and each completion contains 1,526 tokens on average. For the main LLM backbone used in our experiments, `DeepSeek-V3.2`, this corresponds to an average API cost of approximately \$0.22 per dataset. In the default beam-search setting with beam width $K = 3$ and expansion width $m = 3$, each round evaluates up to $K \times m = 9$ candidate programs. These candidate evaluations are independent and can be executed in parallel. Combined with the relatively low latency of TabPFN v2.5 as the search-time evaluator, the practical overhead of multi-trajectory search remains manageable under our experimental setting.

## D. Datasets

In this section, we provide further details on the datasets.

### D.1. Datasets for experiments in Section 3

We summarize the 23 Kaggle datasets used in our main experiments in Table 4. These datasets span binary classification, multi-class classification, and regression, and were selected to cover a wide range of data scales and feature dimensionalities.

**Why Kaggle datasets?** We use Kaggle datasets because the goal of this work is to evaluate semantic LLM-based feature engineering rather than purely anonymous feature construction. FORGE relies on column names, task descriptions, and dataset context to propose and adapt executable feature transformations. Many existing tabular benchmarks are designed primarily for comparing predictive models under standardized preprocessing, and their features are often anonymized, weakly documented, or stripped of the domain context needed for semantic feature generation. In contrast, Kaggle datasets usually provide real-world task descriptions, meaningful column names, and application context, making them suitable for studying whether LLM agents can transfer feature-engineering patterns across semantically described tasks. Importantly, the downstream evaluation datasets are kept disjoint from the SST repository and its near-duplicates, so that the evaluation measures analogical transfer of feature-engineering knowledge rather than direct reuse of dataset-specific solutions. We view evaluation on broader standardized benchmarks as valuable complementary work, especially for studying settings with

weaker semantic metadata, but it is outside the primary scope of this semantic feature-engineering benchmark.

We curate the downstream evaluation benchmark using the following criteria.

**No overlap with the SST repository.** We exclude any dataset that appears in the SST repository, has a near-duplicate or derived version in the SST repository, or is associated with directly corresponding expert solution templates. This filtering prevents direct solution/template reuse through the retrieval channel and ensures that retrieved SST examples provide analogical feature-engineering patterns rather than dataset-specific solutions.

**Leakage-mitigation protocol.** We distinguish *dataset-level exposure*, where an LLM may have seen the dataset name or schema, from *solution-level exposure*, where it may have seen a directly matching high-quality feature-engineering pipeline. Our protocol mainly mitigates the second type: downstream datasets are excluded from the SST repository, near-duplicates are manually filtered, and directly corresponding solution templates are not allowed to be retrieved. We do not claim contamination-free evaluation; instead, we control direct solution/template reuse through the retrieval channel and evaluate analogical transfer of feature-engineering patterns.

**Sufficient number of samples.** We keep only datasets with more than 500 rows. This avoids evaluation being dominated by extremely small-sample instability, where train/validation splits can have high variance and the benefit of feature engineering may be difficult to interpret.

**Clear supervised prediction task.** Each dataset must define an unambiguous target column and a well-specified supervised objective, including binary classification, multi-class classification, or regression. Datasets without a clear target, with ambiguous task definitions, or requiring substantial manual reformulation are excluded.

**Non-saturated raw-feature performance.** We apply a TabPFN-based saturation check to remove tasks that can already be nearly solved using raw features alone. For classification datasets, we treat a task as saturated if raw-feature TabPFN achieves validation accuracy above 99%. For regression datasets, we treat a task as saturated if raw-feature TabPFN achieves validation $R^2$ above 0.99. This check is used only to remove trivially solved tasks, not to select datasets based on FORGE's performance. All downstream evaluation datasets retained in Table 4 pass this check.

**Train/test splitting.** For datasets that provide an official labeled train/test split, we follow the original split whenever available. Otherwise, we use fixed-seed random splits. Candidate feature programs are selected using only the training/validation portion, and final performance is measured on held-out data. All feature transformations are fitted on training data only and then applied to validation/test data to avoid target leakage.

*Table 4.* Summary of datasets in Section 3.

| Dataset Name | Rows | Features | Classes |
|---|---|---|---|
| Rain in Australia | 145,460 | 22 | 2 |
| Telco Customer Churn | 7,043 | 19 | 2 |
| Chocolate Bar Ratings | 1,795 | 8 | - |
| House Sales in King County USA | 21,613 | 20 | - |
| Water Quality | 3,276 | 9 | 2 |
| California Housing Prices | 20,640 | 9 | - |
| Customer Shopping Trends Dataset | 7,800 | 18 | 7 |
| Gender Recognition by Voice | 3,168 | 20 | 2 |
| Apple Quality | 4,000 | 8 | 2 |
| Indicators of Heart Disease (2022) | 1,010,949 | 54 | 2 |
| IBM HR Analytics Employee Attrition | 1,470 | 34 | 2 |
| Goodreads-books | 11,119 | 11 | - |
| Life Expectancy (WHO) | 2,938 | 21 | - |
| Data Science for COVID-19 | 1,643 | 3 | - |
| Diamonds | 53,940 | 10 | - |
| Credit Card customers | 10,127 | 22 | 2 |
| Default of Credit Card Clients Dataset | 30,000 | 24 | 2 |
| Diabetes prediction dataset | 100,000 | 8 | 2 |
| Diabetes Dataset | 768 | 8 | 2 |
| Breast Cancer Wisconsin (Diagnostic) | 569 | 31 | 2 |
| Young People Survey | 1,160 | 151 | 5 |
| Human Activity Recognition | 10,299 | 562 | 6 |
| Forest Fires in Brazil | 6,454 | 4 | - |

## D.2. Datasets for SST

Table 5 lists the Kaggle datasets used to construct our Structured Solution Template (SST) repository. These datasets are distinct from the downstream evaluation benchmark and serve only as sources of expert notebooks from which we extract reusable feature-engineering templates.

*Table 5.* Summary of datasets for SST
(some playground series datasets are not displayed for the convenience of demostration).

| Dataset Name | Rows | Features | Classes |
|---|---|---|---|
| 2024-4-big-data-analytics-certification-kr | 15000 | 9 | 2 |
| airbnb-recruiting-new-user-bookings | 213451 | 30 | 12 |
| allstate-claims-severity | 188318 | 131 | - |
| ashrae-energy-prediction | 19237643 | 15 | - |
| bigquery-geotab-intersection-congestion | 856327 | 27 | - |
| bike-sharing-demand | 10886 | 11 | - |
| bnp-paribas-cardif-claims-management | 114321 | 133 | 2 |
| cat-in-the-dat | 300000 | 23 | 2 |
| chydv-hackathon-2025 | 15000 | 12 | 2 |
| costa-rican-household-poverty-prediction | 9557 | 141 | 4 |
| Customer Personality Analysis | 2240 | 29 | 2 |
| demand-forecasting-kernels-only | 913000 | 3 | - |
| dont-overfit-ii | 250 | 300 | 2 |
| elo-merchant-category-recommendation | 19249694 | 33 | - |
| facebook-recruiting-iv-human-or-bot | 2013 | 11 | 2 |
| favorita-grocery-sales-forecasting | 29118021 | 19 | - |

| | | | |
|---|---|---|---|
| flavours-of-physics | 67553 | 50 | 2 |
| forest-cover-type-kernels-only | 15120 | 54 | 7 |
| ga-customer-revenue-prediction | 54 | 11 | - |
| ghouls-goblins-and-ghosts-boo | 371 | 6 | 3 |
| google-cloud-ncaa-march-madness-2020 | 4502 | 17 | 2 |
| grupo-bimbo-inventory-demand | 74180464 | 13 | - |
| homesite-quote-conversion | 260753 | 2 | 2 |
| how-much-did-it-rain-ii | 1180945 | 23 | - |
| ieee-fraud-detection | 590540 | 394 | 2 |
| instant-gratification | 262144 | 1 | 2 |
| jpx-tokyo-stock-exchange-prediction | 2616876 | 174 | - |
| kobe-bryant-shot-selection | 24557 | 24 | 2 |
| march-machine-learning-mania-2016 | 142482 | 28 | 2 |
| mercedes-benz-greener-manufacturing | 4209 | 1 | - |
| new-york-city-taxi-fare-prediction | 55423856 | 7 | - |
| nyc-taxi-trip-duration | 1458644 | 10 | - |
| optiver-trading-at-the-close | 5237980 | 15 | - |
| otto-group-product-classification-challenge | 61878 | 93 | 9 |
| playground-series-s3e14 | 15289 | 18 | - |
| . . . | . . . | . . . | . . . |
| playground-series-s5e2 | 4394318 | 10 | - |
| porto-seguro-safe-driver-prediction | 595212 | 2 | 2 |
| predict-energy-behavior-of-prosumers | 2014896 | 43 | - |
| predict-student-performance-from-game-play | 424116 | 22 | 2 |
| predict-west-nile-virus | 21012 | 13 | 2 |
| prudential-life-insurance-assessment | 59381 | 12 | 8 |
| pubg-finish-placement-prediction | 4446665 | 28 | - |
| restaurant-revenue-prediction | 100000 | 7 | - |
| rossmann-store-sales | 1017209 | 18 | - |
| santander-customer-satisfaction | 76020 | 369 | 2 |
| santander-customer-transaction-prediction | 200000 | 200 | 2 |
| scrabble-player-rating | 100820 | 20 | - |
| sf-crime | 878049 | 8 | 39 |
| store-sales-time-series-forecasting | 3000888 | 16 | - |
| stumbleupon | 7395 | 26 | 2 |
| tabular-playground-series-apr-2021 | 100000 | 10 | 2 |
| . . . | . . . | . . . | . . . |
| tabular-playground-series-sep-2022 | 70128 | 4 | - |
| telstra-recruiting-network | 7381 | 7 | 3 |
| ventilator-pressure-prediction | 6036000 | 6 | - |

# E. Additional Results

For our ablation study, we selected 3 regression datasets and 3 classification datassets.
**Regression:** Goodreads-books , Forest Fires in Brazil, House Sales in King County USA.
**Classification:** Young People Survey, Customer Shopping Trends Dataset, Diabetes Dataset.

### E.1. Component Ablation

We investigate the impact of retrieval-based knowledge injection (RAG) and beam-style lookahead search by comparing four configurations. Table 6 reports the average relative improvements across three downstream evaluators. Two conclusions are consistent across settings.

**(1) Retrieval and search provide complementary benefits.** Compared with Vanilla CAAFE, adding either RAG or

Beam Search improves performance, indicating that both expert references and multi-trajectory exploration provide useful signals beyond greedy generation. However, their roles are different. RAG alone provides only modest gains because retrieved expert hints still need to be explored and composed through a sufficiently strong search procedure; under a greedy single-trajectory loop, useful retrieved patterns may be under-utilized or discarded early. By contrast, Beam Search directly mitigates search myopia and therefore brings a larger standalone improvement.

**(2) The best results require combining them.** The most informative comparison is Beam Only vs. FORGE (Full). Although Beam Search already improves performance substantially, adding RAG on top of Beam Search further improves the results across downstream evaluators. This suggests that retrieved SST examples are most effective when coupled with lookahead exploration: beam-style search preserves multiple promising trajectories, allowing expert priors to better guide the search toward high-value regions of the feature-program space. Overall, RAG should be viewed as a complementary expert-grounding mechanism that strengthens lookahead search, rather than as a standalone replacement for search.

*Table 6.* **Component ablation study.** Average relative improvement (%). **Beam Search** provides the main gain, while **RAG** adds a further boost only when combined with Beam Search (CAAFE backbone).

| Configuration | TabPFN | Random Forest | XGBoost |
|---|---|---|---|
| Vanilla CAAFE (Baseline) | 1.25% | 0.50% | 0.42% |
| + RAG Only (No Beam) | 1.46% | 1.54% | 1.07% |
| + Beam Only (No RAG) | 2.40% | 2.35% | 2.53% |
| **FORGE (Full)** | **5.52%** | **4.30%** | **3.34%** |

### E.2. The Necessity of Semantic Information.

FORGE relies on semantic metadata (feature names and task descriptions) to propose meaningful transformations. To quantify this effect, we conduct a **Semantic Blindness** test by anonymizing feature names (e.g., replacing "Age" with "Feature_1") and removing dataset descriptions.

Table 7 shows that removing semantic information leads to a substantial performance drop for FORGE across all evaluators (e.g., from **4.41%** to **1.63%** on TabPFN), indicating that semantics plays a key role in guiding LLM-driven feature construction. Notably, the degradation is even more severe for CAAFE: without semantics it can become unstable and even harmful (e.g., **-0.92%** on TabPFN and **-0.12%** on XGBoost), whereas providing feature names alone already recovers consistent gains. These results suggest that semantic context is important for achieving the strongest and most stable gains in agentic AutoFE.

*Table 7.* **Semantic blindness test.** Average relative improvement (%) when removing feature names and dataset descriptions.

| Configuration | TabPFN | Random Forest | XGBoost |
|---|---|---|---|
| FORGE w/ names + descriptions | **4.41%** | **2.02%** | **3.09%** |
| FORGE w/o names + descriptions | 1.63% | 0.62% | 0.74% |
| CAAFE w/ names | 0.40% | 0.46% | 0.51% |
| CAAFE w/o names + descriptions | -0.92% | 0.51% | -0.12% |

### E.3. Effect of Utility Verification in Retrieval.

Beyond retrieving expert references, FORGE further performs utility verification to avoid negative transfer: after task-level retrieval from the SST library, we ask an LLM to assess whether a candidate notebook snippet is *applicable* to the current dataset schema and task. If the verifier predicts the retrieved solution is unlikely to transfer (e.g., relying on missing fields or dataset-specific assumptions), we discard it and do not inject it into the in-context expert reference set $\mathcal{C}$. This procedure corresponds to enabling/disabling the adaptive RAG module described in Section 2.1.

Figure 4 compares FORGE **with** and **without** this rejection mechanism. Adding utility verification yields a slightly higher average relative improvement (**4.94%** vs. **4.59%**), and more importantly, produces **more stable gains** across datasets, as reflected by the reduced variance ($\pm 5.18\%$ vs. $\pm 5.83\%$). These results suggest that adaptive RAG effectively suppresses noisy or mismatched expert references, thereby reducing negative transfer and stabilizing the multi-trajectory search.

*Table 8.* Effect of search-time scoring strategy. We report average relative improvement.

| Search-time scoring strategy | Downstream TabPFN | Downstream XGBoost |
|---|---|---|
| Random/no evaluator | +0.63% | -1.41% |
| TabPFN evaluator | +5.52% | +3.34% |

### E.4. Iteration Dynamics and Convergence

We extend the search horizon to 20 iterations and track the average relative improvement during the search process. Figure 5 shows that FORGE continues to improve over iterations and achieves substantially larger gains than CAAFE, highlighting the benefit of global multi-trajectory search.

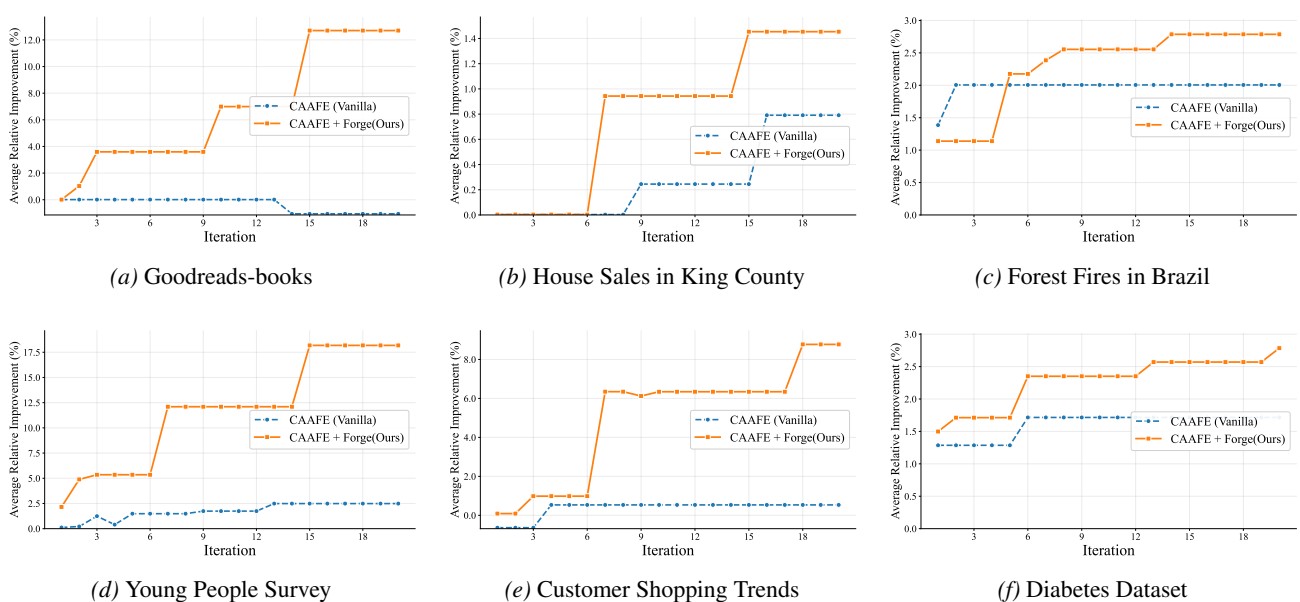

*(a)* Goodreads-books

*(b)* House Sales in King County

*(c)* Forest Fires in Brazil

*(d)* Young People Survey

*(e)* Customer Shopping Trends

*(f)* Diabetes Dataset

*Figure 5.* Detailed comparison of 6 datasets in 20 iterations

### E.5. Impact of the LLM Backbone

To study the sensitivity of LLM-based AutoFE to the choice of backbone model, we instantiate the program generator $\mathcal{M}_\theta$ with four representative LLMs: `Qwen3-Next-80B`, `GPT-4o-mini`, `DeepSeek-V3.2`, and `DeepSeek-V3.2-Exp`. For each backbone, we compare the relative improvement (%) under the same evaluation protocol *with* FORGE and *without* FORGE (the standard generate–execute–evaluate loop).

Figure 6 shows that the backbone choice can noticeably affect the search outcomes, especially in terms of stability across datasets. Without FORGE, the improvements vary substantially across LLMs, indicating that greedy LLM-based refinement is sensitive to the quality of semantic proposals. These results support that FORGE can serve as a plug-and-play enhancement on top of different LLM backbones, rather than relying on a single specific model.

### E.6. Effect of Search-time Validation

We compare TabPFN-based search-time validation against a random/no-evaluator scoring strategy. This sanity check is intended to test whether candidate pruning benefits from validation feedback, rather than to claim that TabPFN is the uniquely optimal proxy evaluator. When evaluator-based validation is removed, LLM-generated features become less reliable and can hurt downstream performance.

As shown in Table 8, using validation feedback from TabPFN yields substantially stronger results than random/no scoring. This supports the need for search-time validation when exploring feature programs, while leaving open the choice of the best proxy under different compute budgets and downstream learners.

*Table 9.* Comparison of TabPFN and XGBoost as search-time evaluators on six datasets. We report average relative improvement and average per-iteration search time.

| Search-time evaluator | Downstream TabPFN | Downstream XGBoost | Time / iter. |
|---|---|---|---|
| TabPFN | +5.52% | +3.34% | 3.22s |
| XGBoost | +1.56% | +4.11% | 21.80s |

*Table 10.* Statistical significance summary on 23 datasets.

| Model | One-sided mean $t$-test $p$-value | Datasets with paired $p < 0.05$ (/23) |
|---|---|---|
| Random Forest | $4.806 \times 10^{-5}$ | 22/23 |
| XGBoost | $4.53 \times 10^{-6}$ | 21/23 |
| TabPFN | $3.446 \times 10^{-5}$ | 22/23 |

## E.7. Choice of Search-time Proxy Evaluator

Our main experiments use TabPFN as the search-time proxy because it provides low-latency candidate ranking without per-candidate retraining or hyperparameter tuning. We do not claim that TabPFN is the best possible evaluator for every downstream learner. To examine this trade-off, we compare TabPFN and XGBoost as search-time evaluators, while evaluating the selected features with both TabPFN and XGBoost downstream.

Table 9 shows a natural evaluator–downstream alignment effect: using XGBoost inside the search loop gives slightly stronger downstream XGBoost performance, whereas TabPFN gives stronger downstream TabPFN performance. However, XGBoost-based search is substantially slower in our setting, requiring 21.80 seconds per iteration on average compared with 3.22 seconds for TabPFN. We therefore use TabPFN as an efficient default proxy that offers a favorable balance between search cost and cross-model usefulness. When the downstream learner is fixed and additional compute is available, replacing TabPFN with the downstream learner itself is a natural extension.

## E.8. Statistical Significance

To assess whether the gains from FORGE are statistically reliable, we performed two complementary one-sided tests on 23 datasets for each downstream model: (i) a one-sided $t$-test on the cross-dataset mean performance, and (ii) per-dataset paired $t$-tests comparing CAAFE and CAAFE+FORGE. The alternative hypothesis is that CAAFE+FORGE outperforms CAAFE.

All three models show highly significant global improvements ($p \ll 0.05$) in the one-sided mean test, and the per-dataset paired tests are also significant on the vast majority of datasets (21–22 out of 23), indicating that the advantage of CAAFE+FORGE is both strong and consistent.

## E.9. Effect of SST Coverage and Semantic Match

We further analyze how the size and semantic match of the SST repository affect FORGE. This analysis is conducted on the 6-dataset ablation subset using TabPFN as the downstream evaluator. Rather than reporting each dataset individually, we summarize the average relative improvement over the no-feature-engineering baseline.

**Effect of SST coverage.** We compare Beam Only with three SST coverage levels: 30%, 70%, and the full SST repository. As shown in Table 11, increasing SST coverage generally improves performance. Beam Only achieves an average relative improvement of 2.40%, while using 70% and 100% of the SST repository improves the average gain to 4.79% and 5.52%, respectively. The 30% setting performs similarly to Beam Only, suggesting that a small or insufficiently diverse SST subset may not provide stable additional benefits. Overall, the trend indicates that SST retrieval is most useful when the repository provides sufficiently broad coverage, allowing the retriever to find semantically relevant expert priors.

**Effect of semantic match.** To further probe whether retrieved knowledge depends on accurate semantic metadata, we compare three task-description settings on the same 6-dataset subset: no dataset description, semantically blurred description, and the original description. The blurred descriptions are rephrased with LLM assistance to preserve only coarse task

*Table 11.* **Effect of SST coverage.** Average relative improvement (%) on the 6-dataset ablation subset using TabPFN as the downstream evaluator.

| Configuration | Beam Only | 30% SST | 70% SST | Full SST |
|---|---|---|---|---|
| Avg. Rel. Improv. | +2.40 | +2.37 | +4.79 | **+5.52** |

information while removing more specific semantic cues. As shown in Table 12, richer and more accurate semantic descriptions lead to stronger gains. The average relative improvement increases from 2.40% with no description to 3.76% with blurred descriptions, and further to 5.52% with the original descriptions. At the same time, FORGE still improves over the no-feature-engineering baseline even without dataset descriptions, suggesting that the method retains practical value when semantic metadata is weak. However, the clear improvement from no description to blurred and original descriptions shows that accurate semantic metadata substantially improves the usefulness of retrieved expert priors.

*Table 12.* **Effect of semantic match.** Average relative improvement (%) on the 6-dataset subset using TabPFN as the downstream evaluator.

| Task-description setting | No description | Blurred description | Original description |
|---|---|---|---|
| Avg. Rel. Improv. | +2.40 | +3.76 | **+5.52** |

These results suggest a nuanced role of retrieval and semantics. SST retrieval is not merely a binary switch: its benefit depends on whether the repository contains sufficiently diverse and semantically relevant templates. Similarly, semantic metadata is not strictly required for FORGE to improve performance, but richer and more accurate metadata makes retrieved expert priors substantially more useful.

### E.10. Budget-Matched Comparison with CAAFE

To further control for search budget, we compare vanilla CAAFE and CAAFE+FORGE under the same number of evaluated candidate programs. Specifically, we extend vanilla CAAFE to 90 sequential generate–execute–evaluate steps and compare it with CAAFE+FORGE, whose default beam setting evaluates up to 90 candidates in total, on a matched subset of 6 datasets. This comparison controls for the number of evaluated feature programs rather than wall-clock time. In practice, FORGE can evaluate the $K \times m$ candidates expanded in each beam-search round in parallel, whereas the 90-step CAAFE baseline follows a sequential greedy trajectory. Thus, this setting is conservative with respect to the practical runtime cost of FORGE.

Table 13 shows that simply increasing the depth of greedy search is not sufficient: even after evaluating the same number of candidate programs, vanilla CAAFE remains consistently below CAAFE+FORGE across all three downstream evaluators.

*Table 13.* **Budget-matched comparison.** Average relative improvement (%) of 90-step vanilla CAAFE versus CAAFE+FORGE under the same number of evaluated candidate programs, evaluated on 6 datasets.

| Downstream evaluator | CAAFE (90 candidates) | CAAFE+FORGE |
|---|---|---|
| Random Forest | +1.11 | **+4.30** |
| TabPFN | +1.92 | **+5.52** |
| XGBoost | +1.43 | **+3.34** |

The gap is substantial for Random Forest (+3.19 points) and TabPFN (+3.60 points), and remains clear for XGBoost (+1.91 points). These results suggest that FORGE's advantage does not merely come from evaluating more candidates, but from improving search quality through multi-trajectory lookahead and expert-guided exploration.

### E.11. Code Example for Beam Search

The generated code in Figure 7 demonstrates the capability of our Beam Search strategy to construct complex, hierarchical features that standard greedy algorithms would likely miss. This process relies on the retention of locally suboptimal candidates to achieve global optimality.

**Preservation of Suboptimal features.** Consider the feature `performance_consistency`. In isolation, this feature

calculated as the inverse distance between MBA and degree performance may exhibit only a moderate correlation with the target variable . A greedy search strategy (Beam Width $B = 1$) would likely discard this feature in favor of simpler, high-correlation metrics like `mba_percentile`. However, our Beam Search ($B = 3$) retains this suboptimal feature in the candidate pool $\mathcal{B}_t$.

**Feature Synthesis.** The value of retaining such features becomes evident in subsequent iterations. As shown in Line 10, `performance_consistency` serves as a critical component for the composite index `career_potential`. This index effectively combines academic rigor with consistency, creating a stronger predictor than its individual parts. By maintaining a diverse beam, the algorithm successfully navigates the dependency.

**Final Ensemble Optimization.** The final feature, `salary_prediction_score`, represents a weighted ensemble of these iteratively refined components. It combines non-linear transformations with interaction terms . This depth of feature engineering is only achievable because the search strategy prioritizes the potential for future combination over immediate standalone performance, effectively escaping local optima in the feature space.

### E.12. Code Example of RAG

In this section , an extract of RAG enhanced feature engineering code is demonstrated, capturing the complex non-linear dynamics inherent in meteorological data beyond standard LLMs.Figure 8 illustrates the core logic.

**Cyclical Encoding.** Temporal variables are transformed into 2D coordinates to preserve their topological continuity, resolving the discontinuity problem where the start and end of a cycle (e.g., Dec 31 and Jan 1) are numerically distant:

$$\mathbf{v}_{time} = \left[ \sin\left(\frac{2\pi t}{T}\right), \cos\left(\frac{2\pi t}{T}\right) \right] \tag{14}$$

**Circular Variable Processing.** For directional variables like wind direction $\theta \in [0, 1)$, standard subtraction fails to account for the wrap-around nature of angles. We compute the shortest geodesic distance on the circle:

$$\Delta\theta = \min(|\theta_{pm} - \theta_{am}|, 1 - |\theta_{pm} - \theta_{am}|) \tag{15}$$

This ensures that small changes crossing the North axis (e.g., $359° \to 2°$) are correctly represented as small differences.

**Physics-Informed Interaction.** Instead of relying solely on the model to infer thermodynamic relationships, we explicitly inject domain knowledge by implementing the Steadman's simplified formula, a standard algorithm employed by the National Weather Service (NWS) for Heat Index (HI) calculation.

$$HI_{simple} = 0.5 \cdot [T + 61.0 + (T - 68.0) \cdot 1.2 + RH \cdot 0.094] \tag{16}$$

This linear approximation serves as the primary evaluation step in the NOAA algorithm. Since our feature target is morning data (Temp9am), which typically falls below the $80°$F threshold required for the complex Rothfusz regression, this simplified formula provides a computationally efficient and physically grounded representation of heat degree without the overfitting risks of high-degree polynomials.

**Short-term Precipitation Momentum.** Meteorological events exhibit strong temporal persistence. To capture the localized trend of rainfall systems while mitigating the stochastic noise of daily measurements, we employ a grouped rolling window aggregation. For each location $l$, the rainfall momentum $M_{l,t}$ at time $t$ is calculated as the moving average over a window $w = 3$:

$$M_{l,t} = \frac{1}{\min(t, w)} \sum_{i=0}^{\min(t,w)-1} R_{l,t-i} \tag{17}$$

where $R_{l,t-i}$ denotes the rainfall at location $l$ with a lag of $i$ days. The grouping operation ensures spatial independence, preventing information leakage across different geographical sites, while the dynamic window size ensures robustness at the temporal boundaries of the dataset.

## F. Limitations and Future Work

**Limitations.** While FORGE consistently improves LLM-based automated feature engineering across diverse Kaggle tabular tasks, the approach relies on the availability of meaningful semantic context (e.g., informative feature names and task

descriptions) to guide program generation; when such metadata is weak or removed, performance can degrade. Moreover, FORGE is designed specifically for tabular prediction tasks, where feature engineering can be expressed as executable transformations over structured columns. Other task formats, such as text, image, graph, sequential decision-making, or multimodal tasks, are outside the scope of the current work.

Another limitation is that the choice of search-time proxy can affect the selected feature programs. We use TabPFN primarily for efficiency and low-latency candidate ranking, not because it is guaranteed to be the optimal evaluator for every downstream model. If the deployment model is fixed and compute is sufficient, using the same model as the search-time evaluator may produce stronger model-specific features, at the cost of substantially higher search time.

**Future Work.** Moving forward, we plan to explore three directions: (1) *Autonomous Library Expansion*: Developing agents that can automatically crawl, validate, and add new high-scoring kernels to the SST library, enabling the system to evolve continuously; (2) *End-to-End Pipeline Integration*: Extending the FORGE logic beyond feature engineering to automate model selection and hyperparameter tuning; and (3) *Human-in-the-Loop Refinement*: Integrating user feedback into the beam search process to interactively steer reasoning branches.

## G. The Use of Large Language Models

We use LLMs as core components of FORGE. Specifically, an LLM is used to generate executable feature-engineering programs, and an LLM-based snippet verifier is used to assess whether retrieved SST snippets are applicable to the current schema before they are injected as in-context references. The downstream validation and final evaluation are performed by tabular predictors such as TabPFN, XGBoost, and Random Forest, rather than by LLMs.

We also used LLMs and generative image models as presentation assistants: LLMs were used to improve grammar and clarity, and image-generation assistance was used to polish the visual layout and aesthetics of the method pipeline figure. The figure content, module definitions, algorithmic flow, experimental design choices, implementation decisions, result analysis, and scientific claims were specified, checked, and verified by the authors.

## H. Acknowledgment of Existing Assets

We gratefully acknowledge the Kaggle platform and the authors of the public notebooks and solutions that serve as sources for constructing our Structured Solution Template (SST) repository. For each SST entry, we retain the original source link to the corresponding Kaggle notebook, competition, or dataset page, and use these materials only in accordance with their publicly available license terms and Kaggle's terms of use. Upon release, we will provide explicit attribution for the original sources and authors whenever such information is available, and will include source links and license/usage information together with the released SST metadata. We will not redistribute any asset whose license or terms prohibit redistribution.

**Prompt**

```
The dataframe `df` is loaded and in memory. Columns are also named attributes.

Description of the dataset in `df` (column dtypes might be inaccurate):
"The collected data has been stored in the Comma Separated Value file Zomato.csv. Each restaurant in the
dataset is uniquely identified by its Restaurant Id. Every Restaurant contains the following variables:"
Columns in `df` (true feature dtypes listed here, categoricals encoded as int):
Restaurant ID (int64): NaN-freq [0.0%], Samples [8417, 301912, 18424175, 18418252, 2958, 18161600, 17534788,
17093600, 5720, 18247005]
Restaurant Name (float64): NaN-freq [0.0%], Samples [1278.4, 1044.2, 1044.2, 1032.49, 1102.38, 464.59, 1036.4,
1032.49, 1421.52, 1083.23]
Country Code (int64): NaN-freq [0.0%], Samples [1, 1, 1, 1, 1, 1, 216, 216, 1, 1]
City (float64): NaN-freq [0.0%], Samples [453.2, 599.58, 544.95, 599.58, 599.58, 544.95, 1036.4, 751.09, 714.37
, 599.58]

...

This code was written by an expert datascientist working to improve predictions. It is a snippet of code that
adds new columns to the dataset.
Number of samples (rows) in training dataset: 7640
Here are some examples of feature engineering code from a similar task, the description of the dataset is:
    Predict annual restaurant sales based on objective measurements,
    which can be used as a reference. You can mimic the knowledge from these examples, but you should not copy
them directly. Instead, generate new code based on the dataset and task description.
    %Example 0:
%Feature Engineering Code:-----------------
from sklearn.experimental import enable_iterative_imputer
from sklearn.impute import IterativeImputer
imp_train = IterativeImputer(max_iter=30, missing_values=0, sample_posterior=True, min_value=1, random_state=37
)
imp_test = IterativeImputer(max_iter=30, missing_values=0, sample_posterior=True, min_value=1, random_state=23)
p_data = ['P'+str(i) for i in range(1,38)]
df[p_data] = np.round(imp_train.fit_transform(df[p_data]))
test_df[p_data] = np.round(imp_test.fit_transform(test_df[p_data]))
columnsToEncode = df.select_dtypes(include=[object]).columns
df = pd.get_dummies(df, columns=columnsToEncode, drop_first=False)
test_df = pd.get_dummies(test_df, columns=columnsToEncode, drop_first=False)
df['revenue'] = np.log1p(df['revenue'])
%Feature Engineering Code 0 End-----------------

Step 1. Analyze the causal relationship or tendency between each feature and task description based on general
knowledge and common sense within a short sentence.
Step 2. Based on the above examples and Step 1's results, give me a code block.
This code generates additional columns that are useful for a downstream classification algorithm (such as
XGBoost) predicting "Average Cost for two".
Additional columns add new semantic information, that is they use real world knowledge on the dataset. They can
 e.g. be feature combinations, transformations, aggregations where the new column is a function of the existing
 columns.
The scale of columns and offset does not matter. Make sure all used columns exist. Follow the above description
 of columns closely and consider the datatypes and meanings of classes.
This code also drops columns, if these may be redundant and hurt the predictive performance of the downstream
classifier (Feature selection). Dropping columns may help as the chance of overfitting is lower, especially if
the dataset is small.
The classifier will be trained on the dataset with the generated columns and evaluated on a holdout set. The
evaluation metric is accuracy. The best performing code will be selected.
Added columns can be used in other codeblocks, dropped columns are not available anymore.

Code formatting for each added column:
```python
# (Feature name and description)
# Usefulness: (Description why this adds useful real world knowledge to classify "Average Cost for two"
according to dataset description and attributes.)
# Input samples: (Three samples of the columns used in the following code, e.g. 'Restaurant ID': [8417, 301912,
 18424175], 'Restaurant Name': [1278.3968076273497, 1044.2015537739537, 1044.2015537739537], ...)
(Some pandas code using Restaurant ID', 'Restaurant Name', ... to add a new column for each row in df)
```end

Code formatting for dropping columns:
```python
# Explanation why the column XX is dropped
df.drop(columns=['XX'], inplace=True)
```end

Each codeblock generates exactly one useful column and can drop unused columns (Feature selection).Each
codeblock ends with ```end and starts with "```python"
Codeblock:
Additional Requirements:1. Any drop operation MUST include a specific column name and justification
2. Never include empty drop operations like df.drop(columns=[''], inplace=True)
3. Always check if a column exists before attempting to drop it
4. Dropping columns should only be suggested when there's clear evidence it would improve model performance
5. Feature additions should always be prioritized over dropping columns6. Never use the {ds[4][-1]} column in
the code, it is the target column, you should not use it to generate new features.
```

*Figure 3.* An example of the prompt in the main program

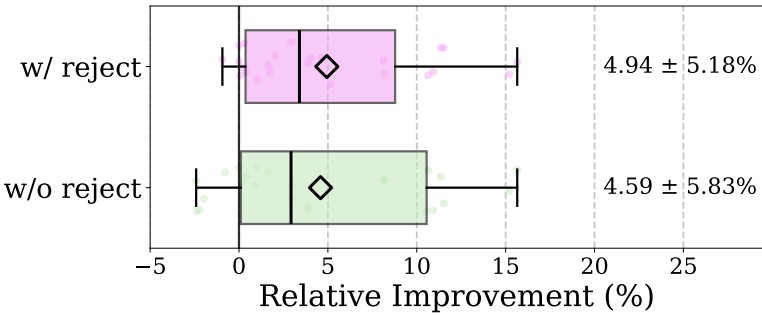

*Figure 4.* Relative improvement (%) of FORGE with and without rejecting non-transferable retrieved expert references. Enabling rejection yields slightly better mean performance and smaller variance, indicating reduced negative transfer.

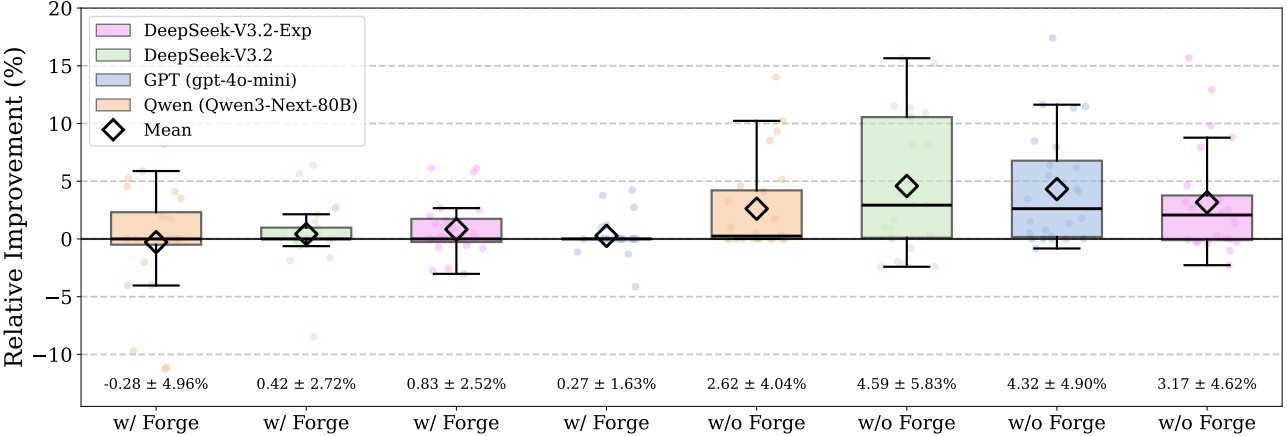

*Figure 6.* **Effect of different LLM backbones.** Relative improvement (%) across four LLM program generators, evaluated with and without FORGE.

```
##### Code generated with Beam Search
# 5. Academic Excellence Score (weighted combination with non-linear terms)
df['academic_excellence'] = (
    np.log1p(df['mba_p']) * 0.45 +
    np.log1p(df['degree_p']) * 0.35 +
    np.log1p(df['etest_p']) * 0.20
)

# 6. Performance Consistency Score (penalizes large gaps)
df['performance_consistency'] = 1 / (1 + abs(df['mba_p'] - df['degree_p']))

# 7. Work Experience Interaction with MBA (non-linear interaction)
df['workex_mba_synergy'] = df['has_workex'] * np.sqrt(df['mba_p'])

# 8. Specialization Performance Multiplier
df['specialization_multiplier'] = df['premium_specialization'] * (df['mba_p'] / 100)

# 9. Composite Percentile Score
df['composite_percentile'] = (df['mba_percentile'] + df['degree_percentile']) / 2

# 10. Career Potential Index (optimized for salary prediction)
df['career_potential'] = (
    df['academic_excellence'] * 0.4 +
    df['has_workex'] * 0.3 +
    df['premium_specialization'] * 0.2 +
    df['performance_consistency'] * 0.1
)

# 11. Salary Prediction Score (optimized weighted combination)
df['salary_prediction_score'] = (
    df['academic_excellence'] * 0.35 +
    df['workex_mba_synergy'] * 0.25 +
    df['career_potential'] * 0.20 +
    df['specialization_multiplier'] * 0.15 +
    df['composite_percentile'] * 0.05
)

-------------------------------------------------------------------

##### Code generated by CAAFE
# Academic consistency score: Combines performance across education levels (ssc, hsc,
    degree, mba) into a single weighted average.
# Usefulness: Salary often correlates with consistent academic performance across
    educational stages. This feature captures overall academic excellence.
# Input samples: 'ssc_p': [69.0, 80.92, 62.0], 'hsc_p': [65.0, 78.5, 62.0], 'degree_p':
    [57.0, 67.0, 60.0], 'mba_p': [55.8, 60.99, 52.38]
df['academic_consistency'] = (df['ssc_p'] * 0.2 + df['hsc_p'] * 0.25 + df['degree_p'] *
    0.3 + df['mba_p'] * 0.25)
# Work experience premium: Interaction between work experience and MBA performance.
# Usefulness: Candidates with work experience and high MBA scores may command higher
    salaries. This feature captures that synergy.
# Input samples: 'workex': [273139.37, 273139.37, 304178.87], 'mba_p': [55.8, 60.99,
    52.38]
df['workex_mba_interaction'] = df['workex'] * df['mba_p']
```

*Figure 7.* Illustrative code snippets produced by beam search (top) vs. a greedy baseline (CAAFE) (bottom) on the salary prediction task. The beam-search trajectory composes multi-step, hierarchical features that enable downstream combinations.

```
###### Code generated with RAG
# 1. Topological Preservation of Seasonality
df['Seasonal_Sine'] = np.sin(2 * np.pi * df['Date'])
df['Seasonal_Cosine'] = np.cos(2 * np.pi * df['Date'])

# 2. Geodesic Distance for Circular Variables (Wind Direction)
df['wind_dir_diff'] = np.abs(df['WindDir3pm'] - df['WindDir9am'])
df['wind_dir_diff'] = np.minimum(df['wind_dir_diff'], 1 - df['wind_dir_diff'])

# 3. Spatio-Temporal Anomaly Detection
# Subtract historical seasonal mean to isolate current anomalies
seasonal_mean = df.groupby(['Location', 'DayOfYear'])['Humidity3pm'].transform('mean')
df['Humidity_Anomaly'] = df['Humidity3pm'] - seasonal_mean

# 4. Injection of Physical Empirical Formulas (Steadman's Simplified Formula)
df['heat_index'] = 0.5 * (df['Temp'] + 61.0 + (df['Temp'] - 68.0) * 1.2 + df['Humidity'] *
    0.094)

#5.Group Sliding Window
df['Rainfall_Momentum_3day'] = df.groupby('Location')['Rainfall'].transform(lambda x: x.
    rolling(3, min_periods=1).mean())

----------------------------------------------------------------------------

##### Code generated with CAAFE
df['Season'] = df['Month'].apply(lambda x: 'Summer' if x in [12, 1, 2] else
'Autumn' if x in [3, 4, 5] else
'Winter' if x in [6, 7, 8] else 'Spring')
# Temperature range and stability features
# Usefulness: Large temperature swings and unstable temperature patterns often precede
    precipitation events
# Input samples: 'MinTemp': [14.5, 24.6, 14.7], 'MaxTemp': [17.8, 39.5, 26.6], 'Temp9am':
    [16.2, 30.1, 17.3], 'Temp3pm': [21.68, 39.2, 24.5]
df['TempRange'] = df['MaxTemp'] - df['MinTemp']
df['TempMorningRise'] = df['Temp9am'] - df['MinTemp']
df['TempAfternoonRise'] = df['Temp3pm'] - df['Temp9am']
df['TempStability'] = (df['TempAfternoonRise'] / (df['TempRange'] + 1e-6)).abs()  #
    Normalized stability measure
```

*Figure 8.* Illustrative code snippets produced by RAG-enhanced generation (top) vs. CAAFE (bottom) on a weather prediction task. RAG introduces domain-informed transformations (e.g., seasonality, circular wind direction, and grouped rolling statistics) that go beyond generic heuristics.

