# OpenReview forum: "Lookahead Automated Feature Engineering for Tabular Prediction via Kaggle-Guided Knowledge Transfer"
_ICML.cc/2026/Workshop/FMSD — FMSD @ ICML 2026 Poster_

### Official Review · Reviewer_L2Qm · 2026-05-21
**Engineering toolbox for LLM-based feature engineering**

**Rating:** 5
**Confidence:** 4

**Review:**

The authors propose FORGE a retrieval-augmented "lookahead" framework for LLM-based feature engineering on tabular data.
The method is designed to target two failure modes of existing LLM-based methods (e.g., CAAFE, OCTree):
"trivialization of expertise" (LLMs propose generic safe transformations); and "myopia of stepwise search" (greedy generate-evaluate-refine loops discard intermediate features whose utility only emerges after composition).
FORGE addresses these via three components: a repository of Structured Solution Templates (SSTs) distilled from top-3 solutions across 90 Kaggle competitions, retrieved by task similarity and filtered by an LLM-based snippet-level utility verifier; a tree-structured beam search over feature transformations; TabPFN as a low-latency search-time proxy evaluator.
In experiments they use TabPFN, XGBoost, and Random Forest on 23 Kaggle datasets.
FORGE improves over CAAFE and OCTree backbones in average rank, average relative improvement and per-dataset win counts.
Ablations show that most gains come from beam search, with RAG over the SSTs providing light additional improvements.

The paper identifies practical failure modes of existing LLM-based feature engineering methods and the framing of "trivialization" and "myopia" is sensible and well-motivated. The SST repository seems to be well engineered with templates from 90 top Kaggle solutions. The component ablation tries to disentangle the contributions of RAG and beam search. Authors acknowledge that TabPFN may not be the optimal proxy for all downstream learners, and that semantic metadata is a hard requirement.

On the flip-side, I believe it is a miss to not include LLM-FE (Abhyankar et al. 2025) as a baseline. It is cited but the authors do not compare against it experimentally. LLM-FE directly targets the same myopia problem via population-based evolutionary search and it is the closest competitor in the design space (LLM + program search + multi-trajectory exploration).
From a methodological contribution, all three components in FORGE are more or less off-the-shelf (RAG, beam search over LLM output, TabPFN as a fast tabular evaluator). The novel contributions are essentially: the SST library, snippet-level LLM-based utility verification, the empirical demonstration that the combination works reasonably well. On most datasets median improvements are relatively small though and practical gains are little (0.4-0.8% across evaluators). There is no comparison to lighter non-greedy alternatives such as: best-of-N sampling at each step, random restarts, or a simple population without the structured beam. This would help isolate whether structured lookahead is needed, or whether any diversification mechanism is enough (also related to not comparing to LLM-FE).

Questions / Comments:
* Table 1: Please include standard deviations or confidence intervals. A single mean across 23 datasets is hard to interpret.
* It might be interesting to evaluate on standard non-Kaggle benchmarks (OpenML-CC18, TabZilla, ...). Even if FORGE underperforms there due to weak metadata, it would better characterize the method's domain.
* To increase relevancy to the workshop it would be interesting to evaluate more TFMs than only TabPFN (for both downstream evaluators and fast evaluators within FORGE).

Overall, the paper appears as a solid engineering contribution but it is unclear how it would compare to other LLM-based methods that include diversity mechanisms in their search. In general, improvements from FORGE are somewhat incremental on most tasks. Given these results and limited methodological contribution it is difficult to actively vote for acceptance.

---

### Official Review · Reviewer_nWA8 · 2026-05-22
**Thorough and Practical Multi-Trajectory Search for Tabular Feature Engineering**

**Rating:** 9
**Confidence:** 3

**Review:**

### **Summary**

The paper introduces FORGE (Feature Optimization with Retrieved knowledge Guidance and lookahead Exploration), a framework designed to enhance Large Language Model (LLM) based Automated Feature Engineering (AutoFE) for tabular data tasks. The authors target two key drawbacks of present agentic approaches: "expertise trivialization" (where LLMs default to shallow, generic operations) and "search myopia" (where greedy, sequential selection strategies prematurely drop valuable intermediate features).

To address these challenges, FORGE introduces a three-tier design:
- Structured Knowledge Injection: Extracts self-contained computational logic from top-performing historical Kaggle solution notebooks into a Structured Solution Template (SST) library, injecting verified, task-relevant patterns as in-context guidance.
- Multi-Trajectory Program Search: Implements a tree-structured beam search over candidate feature programs to safely retain locally suboptimal features whose synergistic utility only emerges in later multi-step compositions.
- Efficient Search-Time Proxy: Minimizes candidate evaluation overhead by leveraging TabPFN as a training-free proxy to provide low-latency relative ranking rewards.

### **Strengths**
- **Smart Search Method:** Instead of using a basic greedy loop that only picks one option at a time, this framework lets the model explore multiple paths at once using a beam search. This stops the system from being too short-sighted. It keeps intermediate steps active because even if a feature looks weak on its own, it might work perfectly when combined with other features later on.

- **Fast Evaluation:** Testing all these different feature combinations usually takes a ton of time and compute. By using TabPFN as a quick proxy evaluator, the authors cut down the testing time per round from about 22 seconds with XGBoost to just over 3 seconds. This makes looking ahead on multiple paths actually practical.

- **Thorough Validation and Sanity Checks:** The authors include a robust set of extra experiments to validate their approach. First, they ran a "budget-matched" test where the standard greedy baseline was extended to evaluate the exact same number of candidate features. This proves that FORGE wins due to its lookahead search strategy rather than just looking at more options. Second, they ran a "semantic blindness" test by anonymizing column names and removing task descriptions. The significant drop in performance during this test clearly demonstrates that the framework successfully utilizes column meanings to build smarter features.

- **Solid Results:** Testing the framework on 23 different datasets with strict statistical tests (like paired t-tests) makes the final results very trustworthy. It shows the performance boost is consistent and isn't just a fluke on a couple of lucky tables.

### **Areas for Improvement**
- **Clarity on Baseline Integration:** The paper heavily features combinations like "FORGE + CAAFE" and "FORGE + OCTree" in its main benchmarks. However, the exact prompt-level or code-level integration between FORGE and these baselines is quite vague in the main text. It would greatly improve the paper if the authors explicitly explained how FORGE intercepts the baseline loops, specifically how it wraps their unique prompting styles into its own multi-trajectory search equation.

- **Automation of the SST Library Pipeline:** The authors describe a standardization pipeline that separates reusable feature engineering code from dataset-specific rules inside Kaggle notebooks. However, they do not explain how automated this parser actually is.

- **Heavy Reliance on Clean Metadata:** The framework is heavily dependent on clean, natural-language feature names and task descriptions to function. As shown in the "semantic blindness" test, stripping away this text metadata causes a major drop in performance across all downstream evaluators. This raises a question how large of a problem it is in real use cases, elevated from well curated Kaggle style datasets.

### **Detailed Comments**
- **Proxy Model Bias:** The data shows an alignment effect where using XGBoost during the search loop gives better downstream XGBoost results than using TabPFN. The authors could further discuss whether using a transformer proxy like TabPFN biases the engineered features away from what is actually best for tree-based models.

- Could you clarify how automated the creation of the SST library is? Does it require manual human filtering to extract clean feature engineering templates from the Kaggle notebooks?

- If a dataset has completely cryptic or missing column names, does the retrieval step have a way to fall back on structural similarity, or does it risk pulling irrelevant code templates?

### **Justification of Score**
**Score: strong accept**

The paper presents a solid, well-motivated solution to clear flaws in current LLM-based feature engineering agents. The combination of expert template retrieval and lookahead beam search is clever , and using TabPFN as a cheap proxy keeps the system fast and practical. Backed by thorough ablation experiments and consistent statistical wins across 23 datasets , this work is a strong fit for the workshop.

---

### Official Review · Reviewer_hhdj · 2026-05-22
**Review for ICML 2026 Workshop FMSD Submission79.**

**Rating:** 8
**Confidence:** 4

**Review:**

- Summary: The paper introduces FORGE, a retrieval-augmented framework that uses LLM for feature engineering. It uses Kaggle solutions and optimizes several factors to make FORGE effective and efficient.

- Strengths: Overall, the paper is easy to follow. The experimental results provide the strength of FORGE, and the paper provides detailed information through the appendix.

- Comments: If possible, it would be interesting to compare with the 'TabPrep' functionality from the Autogluon package, which is also a feature-engineering framework. Figure 2 can be resized for better readability. Some question arise: how are the categorical or non-numerical features handled? When tabular data contains textual information or non-numeric features (which is indeed a case for many of the real-world datasets) how can FORGE be extended? What are some potential extensions of FORGE for relational data?

- Justification of Score: Overall, the paper is well-structured with detailed information. I would be interesting to learn a bit more on the prospective of the related approaches for feature engineering on structured data with LLMs.